# Man is to Computer Programmer as Woman is to Homemaker? Debiasing Word Embeddings

**Tolga Bolukbasi**[1], **Kai-Wei Chang**[2], **James Zou**[2], **Venkatesh Saligrama**[1,2], **Adam Kalai**[2]
[1]Boston University, 8 Saint Mary's Street, Boston, MA
[2]Microsoft Research New England, 1 Memorial Drive, Cambridge, MA
tolgab@bu.edu, kw@kwchang.net, jamesyzou@gmail.com, srv@bu.edu, adam.kalai@microsoft.com

## Abstract

The blind application of machine learning runs the risk of amplifying biases present in data. Such a danger is facing us with *word embedding*, a popular framework to represent text data as vectors which has been used in many machine learning and natural language processing tasks. We show that even word embeddings trained on Google News articles exhibit female/male gender stereotypes to a disturbing extent. This raises concerns because their widespread use, as we describe, often tends to amplify these biases. Geometrically, gender bias is first shown to be captured by a direction in the word embedding. Second, gender neutral words are shown to be linearly separable from gender definition words in the word embedding. Using these properties, we provide a methodology for modifying an embedding to remove gender stereotypes, such as the association between the words *receptionist* and *female*, while maintaining desired associations such as between the words *queen* and *female*. Using crowd-worker evaluation as well as standard benchmarks, we empirically demonstrate that our algorithms significantly reduce gender bias in embeddings while preserving the its useful properties such as the ability to cluster related concepts and to solve analogy tasks. The resulting embeddings can be used in applications without amplifying gender bias.

## 1 Introduction

Research on word embeddings has drawn significant interest in machine learning and natural language processing. There have been hundreds of papers written about word embeddings and their applications, from Web search [22] to parsing Curriculum Vitae [12]. However, none of these papers have recognized how blatantly sexist the embeddings are and hence risk introducing biases of various types into real-world systems.

A word embedding, trained on word co-occurrence in text corpora, represents each word (or common phrase) $w$ as a $d$-dimensional *word vector* $\vec{w} \in \mathbb{R}^d$. It serves as a dictionary of sorts for computer programs that would like to use word meaning. First, words with similar semantic meanings tend to have vectors that are close together. Second, the vector differences between words in embeddings have been shown to represent relationships between words [27, 21]. For example given an analogy puzzle, "man is to king as woman is to $x$" (denoted as *man*:*king* :: *woman*:$x$), simple arithmetic of the embedding vectors finds that $x$=*queen* is the best answer because $\overrightarrow{man} - \overrightarrow{woman} \approx \overrightarrow{king} - \overrightarrow{queen}$. Similarly, $x$=*Japan* is returned for *Paris*:*France* :: *Tokyo*:$x$. It is surprising that a simple vector arithmetic can simultaneously capture a variety of relationships. It has also excited practitioners because such a tool could be useful across applications involving natural language. Indeed, they are being studied and used in a variety of downstream applications (e.g., document ranking [22], sentiment analysis [14], and question retrieval [17]).

However, the embeddings also pinpoint sexism implicit in text. For instance, it is also the case that:

$$\overrightarrow{man} - \overrightarrow{woman} \approx \overrightarrow{computer\ programmer} - \overrightarrow{homemaker}.$$

In other words, the same system that solved the above reasonable analogies will offensively answer "man is to computer programmer as woman is to $x$" with $x$=*homemaker*. Similarly, it outputs that a

| Extreme *she* | Extreme *he* | Gender stereotype *she-he* analogies | | |
|---|---|---|---|---|
| 1. homemaker | 1. maestro | | | |
| 2. nurse | 2. skipper | sewing-carpentry | registered nurse-physician | housewife-shopkeeper |
| 3. receptionist | 3. protege | nurse-surgeon | interior designer-architect | softball-baseball |
| 4. librarian | 4. philosopher | blond-burly | feminism-conservatism | cosmetics-pharmaceuticals |
| 5. socialite | 5. captain | giggle-chuckle | vocalist-guitarist | petite-lanky |
| 6. hairdresser | 6. architect | sassy-snappy | diva-superstar | charming-affable |
| 7. nanny | 7. financier | volleyball-football | cupcakes-pizzas | lovely-brilliant |
| 8. bookkeeper | 8. warrior | Gender appropriate *she-he* analogies | | |
| 9. stylist | 9. broadcaster | queen-king | sister-brother | mother-father |
| 10. housekeeper | 10. magician | waitress-waiter | ovarian cancer-prostate cancer | convent-monastery |

Figure 1: **Left** The most extreme occupations as projected on to the *she−he* gender direction on w2vNEWS. Occupations such as *businesswoman*, where gender is suggested by the orthography, were excluded. **Right** Automatically generated analogies for the pair *she-he* using the procedure described in text. Each automatically generated analogy is evaluated by 10 crowd-workers to whether or not it reflects gender stereotype.

*father* is to a *doctor* as a *mother* is to a *nurse*. The primary embedding studied in this paper is the popular publicly-available word2vec [19, 20] 300 dimensional embedding trained on a corpus of Google News texts consisting of 3 million English words, which we refer to here as the w2vNEWS. One might have hoped that the Google News embedding would exhibit little gender bias because many of its authors are professional journalists. We also analyze other publicly available embeddings trained via other algorithms and find similar biases (Appendix B).

In this paper, we quantitatively demonstrate that word-embeddings contain biases in their geometry that reflect gender stereotypes present in broader society.[1] Due to their wide-spread usage as basic features, word embeddings not only reflect such stereotypes but can also amplify them. This poses a significant risk and challenge for machine learning and its applications. The analogies generated from these embeddings spell out the bias implicit in the data on which they were trained. Hence, word embeddings may serve as a means to extract implicit gender associations from a large text corpus similar to how Implicit Association Tests [11] detect automatic gender associations possessed by people, which often do not align with self reports.

To quantify bias, we will compare a word vector to the vectors of a pair of gender-specific words. For instance, the fact that $\overrightarrow{nurse}$ is close to $\overrightarrow{woman}$ is not in itself necessarily biased(it is also somewhat close to $\overrightarrow{man}$ – all are humans), but the fact that these distances are unequal suggests bias. To make this rigorous, consider the distinction between *gender specific* words that are associated with a gender by definition, and the remaining *gender neutral* words. Standard examples of gender specific words include *brother*, *sister*, *businessman* and *businesswoman*. We will use the gender specific words to learn a gender subspace in the embedding, and our debiasing algorithm removes the bias only from the gender neutral words while respecting the definitions of these gender specific words.

We propose approaches to reduce gender biases in the word embedding while preserving the useful properties of the embedding. Surprisingly, not only does the embedding capture bias, but it also contains sufficient information to reduce this bias.We will leverage the fact that there exists a low dimensional subspace in the embedding that empirically captures much of the gender bias.

## 2 Related work and Preliminary

**Gender bias and stereotype in English.** It is important to quantify and understand bias in languages as such biases can reinforce the psychological status of different groups [28]. Gender bias in language has been studied over a number of decades in a variety of contexts (see, e.g., [13]) and we only highlight some of the findings here. Biases differ across people though commonalities can be detected. Implicit Association Tests [11] have uncovered gender-word biases that people do not self-report and may not even be aware of. Common biases link female terms with liberal arts and family and male terms with science and careers [23]. Bias is seen in word morphology, i.e., the fact that words such as

*actor* are, by default, associated with the dominant class [15], and female versions of these words, e.g., *actress*, are marked. There is also an imbalance in the number of words with F-M with various associations. For instance, while there are more words referring to males, there are many more words that sexualize females than males [30]. Consistent biases have been studied within online contexts and specifically related to the contexts we study such as online news (e.g., [26]), Web search (e.g., [16]), and Wikipedia (e.g., [34]).

**Bias within algorithms.** A number of online systems have been shown to exhibit various biases, such as racial discrimination and gender bias in the ads presented to users [31, 4]. A recent study found that algorithms used to predict repeat offenders exhibit indirect racial biases [1]. Different demographic and geographic groups also use different dialects and word-choices in social media [6]. An implication of this effect is that language used by minority group might not be able to be processed by natural language tools that are trained on "standard" data-sets. Biases in the curation of machine learning data-sets have explored in [32, 3].

Independent from our work, Schmidt [29] identified the bias present in word embeddings and proposed debiasing by entirely removing multiple gender dimensions, one for each gender pair. His goal and approach, similar but simpler than ours, was to entirely remove gender from the embedding. There is also an intense research agenda focused on improving the quality of word embeddings from different angles (e.g., [18, 25, 35, 7]), and the difficulty of evaluating embedding quality (as compared to supervised learning) parallels the difficulty of defining bias in an embedding.

Within machine learning, a body of notable work has focused on "fair" binary classification in particular. A definition of fairness based on legal traditions is presented by Barocas and Selbst [2]. Approaches to modify classification algorithms to define and achieve various notions of fairness have been described in a number of works, see, e.g., [2, 5, 8] and a recent survey [36]. The prior work on algorithmic fairness is largely for supervised learning. Fair classification is defined based on the fact that algorithms were classifying a set of individuals using a set of features with a distinguished sensitive feature. In word embeddings, there are no clear individuals and no a priori defined classification problem. However, similar issues arise, such as direct and indirect bias [24].

**Word embedding.** An embedding consists of a unit vector $\vec{w} \in \mathbb{R}^d$, with $\|\vec{w}\| = 1$, for each word (or term) $w \in W$. We assume there is a set of gender neutral words $N \subset W$, such as *flight attendant* or *shoes*, which, by definition, are not specific to any gender. We denote the size of a set $S$ by $|S|$. We also assume we are given a set of F-M gender pairs $P \subset W \times W$, such as *she-he* or *mother-father* whose definitions differ mainly in gender. Section 5 discusses how $N$ and $P$ can be found within the embedding itself, but until then we take them as given. As is common, *similarity* between two vectors $u$ and $v$ can be measured by their *cosine similarity* : $\cos(u, v) = \frac{u \cdot v}{\|u\|\|v\|}$. This normalized similarity between vectors $u$ and $v$ is the cosine of the angle between the two vectors. Since words are normalized $\cos(\vec{w}_1, \vec{w}_2) = \vec{w}_1 \cdot \vec{w}_2$.[2]

Unless otherwise stated, the embedding we refer to is the aforementioned w2vNEWS embedding, a $d = 300$-dimensional word2vec [19, 20] embedding, which has proven to be immensely useful since it is high quality, publicly available, and easy to incorporate into any application. In particular, we downloaded the pre-trained embedding on the Google News corpus,[3] and normalized each word to unit length as is common. Starting with the 50,000 most frequent words, we selected only lower-case words and phrases consisting of fewer than 20 lower-case characters (words with upper-case letters, digits, or punctuation were discarded). After this filtering, 26,377 words remained. While we focus on w2vNEWS, we show later that gender stereotypes are also present in other embedding data-sets. **Crowd experiments.**[4] Two types of experiments were performed: ones where we solicited words from the crowd (to see if the embedding biases contain those of the crowd) and ones where we solicited ratings on words or analogies generated from our embedding (to see if the crowd's biases contain those from the embedding). These two types of experiments are analogous to experiments performed in rating results in information retrieval to evaluate precision and recall. When we speak of the majority of 10 crowd judgments, we mean those annotations made by 5 or more independent workers. The Appendix contains the questionnaires that were given to the crowd-workers.

# 3 Geometry of Gender and Bias in Word Embeddings

Our first task is to understand the biases present in the word-embedding (i.e. which words are closer to *she* than to *he*, etc.) and the extent to which these geometric biases agree with human notion of gender stereotypes. We use two simple methods to approach this problem: 1) evaluate whether the embedding has stereotypes on occupation words and 2) evaluate whether the embedding produces analogies that are judged to reflect stereotypes by humans. The exploratory analysis of this section will motivate the more rigorous metrics used in the next two sections.

**Occupational stereotypes.** Figure 1 lists the occupations that are closest to *she* and to *he* in the w2vNEWS embeddings. We asked the crowdworkers to evaluate whether an occupation is considered female-stereotypic, male-stereotypic, or neutral. The projection of the occupation words onto the *she-he* axis is strongly correlated with the stereotypicality estimates of these words (Spearman $\rho = 0.51$), suggesting that the geometric biases of embedding vectors is aligned with crowd judgment. We projected each of the occupations onto the *she-he* direction in the w2vNEWS embedding as well as a different embedding generated by the GloVe algorithm on a web-crawl corpus [25]. The results are highly consistent (Appendix Figure 6), suggesting that gender stereotypes is prevalent across different embeddings and is not an artifact of the particular training corpus or methodology of word2vec.

**Analogies exhibiting stereotypes.** Analogies are a useful way to both evaluate the quality of a word embedding and also its stereotypes. We first briefly describe how the embedding generate analogies and then discuss how we use analogies to quantify gender stereotype in the embedding. A more detailed discussion of our algorithm and prior analogy solvers is given in Appendix C.

In the standard analogy tasks, we are given three words, for example *he, she, king*, and look for the 4th word to solve *he* to *king* is as *she* to $x$. Here we modify the analogy task so that given two words, e.g. *he, she*, we want to generate a pair of words, $x$ and $y$, such that *he* to $x$ as *she* to $y$ is a good analogy. This modification allows us to systematically generate pairs of words that the embedding believes it analogous to *he, she* (or any other pair of seed words). The input into our analogy generator is a seed pair of words $(a, b)$ determining a *seed direction* $\vec{a} - \vec{b}$ corresponding to the normalized difference between the two seed words. In the task below, we use $(a, b) = (\text{she}, \text{he})$. We then score all pairs of words $x, y$ by the following metric:

$$S_{(a,b)}(x, y) = \cos\left(\vec{a} - \vec{b}, \vec{x} - \vec{y}\right) \text{ if } \|\vec{x} - \vec{y}\| \leq \delta, \quad 0 \text{ else} \tag{1}$$

where $\delta$ is a threshold for similarity. The intuition of the scoring metric is that we want a good analogy pair to be close to parallel to the seed direction while the two words are not too far apart in order to be semantically coherent. The parameter $\delta$ sets the threshold for semantic similarity. In all the experiments, we take $\delta = 1$ as we find that this choice often works well in practice. Since all embeddings are normalized, this threshold corresponds to an angle $\leq \pi/3$, indicating that the two words are closer to each other than they are to the origin. In practice, it means that the two words forming the analogy are significantly closer together than two random embedding vectors. Given the embedding and seed words, we output the top analogous pairs with the largest positive $S_{(a,b)}$ scores. To reduce redundancy, we do not output multiple analogies sharing the same word $x$.

We employed U.S. based crowd-workers to evaluate the analogies output by the aforementioned algorithm. For each analogy, we asked the workers two yes/no questions: (a) whether the pairing makes sense as an analogy, and (b) whether it reflects a gender stereotype. Overall, 72 out of 150 analogies were rated as gender-appropriate by five or more out of 10 crowd-workers, and 29 analogies were rated as exhibiting gender stereotype by five or more crowd-workers (Figure 4). Examples of analogies generated from w2vNEWS are shown at Figure 1. The full list are in Appendix J.

**Identifying the gender subspace.** Next, we study the bias present in the embedding geometrically, identifying the gender direction and quantifying the bias independent of the extent to which it is aligned with the crowd bias. Language use is "messy" and therefore individual word pairs do not always behave as expected. For instance, the word *man* has several different usages: it may be used as an exclamation as in *oh man!* or to refer to people of either gender or as a verb, e.g., *man the station*. To more robustly estimate bias, we shall aggregate across multiple paired comparisons. By combining several directions, such as $\overrightarrow{\text{she}} - \overrightarrow{\text{he}}$ and $\overrightarrow{\text{woman}} - \overrightarrow{\text{man}}$, we identify a **gender direction** $g \in \mathbb{R}^d$ that largely captures gender in the embedding. This direction helps us to quantify direct and indirect biases in words and associations.

In English as in many languages, there are numerous gender pair terms, and for each we can consider the difference between their embeddings. Before looking at the data, one might imagine

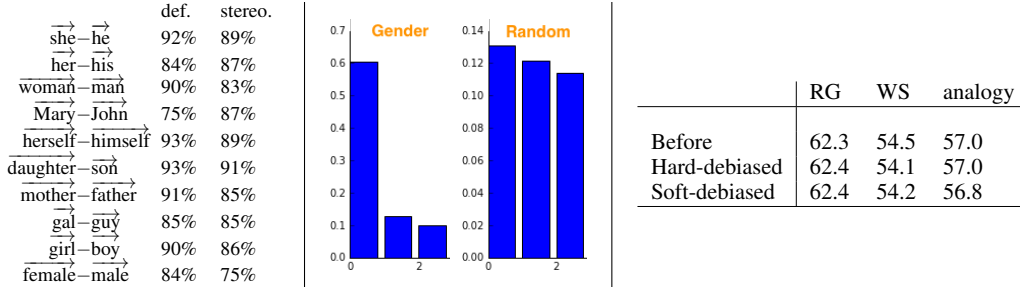

Figure 2: **Left:** Ten word pairs to define gender, along with agreement with sets of definitional and stereotypical words solicited from the crowd. The accuracy is shown for the corresponding gender classifier based on which word is closer to a target word, e.g., the *she-he* classifier predicts a word is female if it is closer to *she* than *he*. **Middle:** The bar plot shows the percentage of variance explained in the PCA of the 10 pairs of gender words. The top component explains significantly more variance than any other; the corresponding percentages for random words shows a more gradual decay (Figure created by averaging over 1,000 draws of ten random unit vectors in 300 dimensions). **Right:** The table shows performance of the original w2vNEWS embedding ("before") and the debiased w2vNEWS on standard evaluation metrics measuring coherence and analogy-solving abilities: RG [27], WS [10], MSR-analogy [21]. Higher is better. The results show that the performance does not degrade after debiasing. Note that we use a subset of vocabulary in the experiments. Therefore, the performances are lower than the previously published results. See Appendix for full results.

that they all had roughly the same vector differences, as in the following caricature: $\overrightarrow{\text{grandmother}} = \overrightarrow{\text{wise}} + \overrightarrow{\text{gal}}$, $\overrightarrow{\text{grandfather}} = \overrightarrow{\text{wise}} + \overrightarrow{\text{guy}}$, $\overrightarrow{\text{grandmother}} - \overrightarrow{\text{grandfather}} = \overrightarrow{\text{gal}} - \overrightarrow{\text{guy}} = g$ However, gender pair differences are not parallel in practice, for multiple reasons. First, there are different biases associated with with different gender pairs. Second is polysemy, as mentioned, which in this case occurs due to the other use of *grandfather* as in *to grandfather a regulation*. Finally, randomness in the word counts in any finite sample will also lead to differences. Figure 2 illustrates ten possible gender pairs, $\left\{(x_i, y_i)\right\}_{i=1}^{10}$.

To identify the gender subspace, we took the ten gender pair difference vectors and computed its principal components (PCs). As Figure 2 shows, there is a single direction that explains the majority of variance in these vectors. The first eigenvalue is significantly larger than the rest. Note that, from the randomness in a finite sample of ten noisy vectors, one expects a decrease in eigenvalues. However, as also illustrated in 2, the decrease one observes due to random sampling is much more gradual and uniform. Therefore we hypothesize that the top PC, denoted by the unit vector $g$, captures the gender subspace. In general, the gender subspace could be higher dimensional and all of our analysis and algorithms (described below) work with general subspaces.

**Direct bias.** To measure direct bias, we first identify words that should be gender-neutral for the application in question. How to generate this set of gender-neutral words is described in Section 5. Given the gender neutral words, denoted by $N$, and the gender direction learned from above, $g$, we define the direct gender bias of an embedding to be $\frac{1}{|N|} \sum_{w \in N} |\cos(\vec{w}, g)|^c$, where $c$ is a parameter that determines how *strict* do we want to in measuring bias. If $c$ is 0, then $|\cos(\vec{w} - g)|^c = 0$ only if $\vec{w}$ has no overlap with $g$ and otherwise it is 1. Such strict measurement of bias might be desirable in settings such as the college admissions example from the Introduction, where it would be unacceptable for the embedding to introduce a slight preference for one candidate over another by gender. A more gradual bias would be setting $c = 1$. The presentation we have chosen favors simplicity – it would be natural to extend our definitions to weight words by frequency. For example, in w2vNEWS, if we take $N$ to be the set of 327 occupations, then $\text{DirectBias}_1 = 0.08$, which confirms that many occupation words have substantial component along the gender direction.

## 4 Debiasing algorithms

The debiasing algorithms are defined in terms of sets of words rather than just pairs, for generality, so that we can consider other biases such as racial or religious biases. We also assume that we have a set of words to neutralize, which can come from a list or from the embedding as described in Section 5. (In many cases it may be easier to list the gender specific words not to neutralize as this set can be much smaller.)

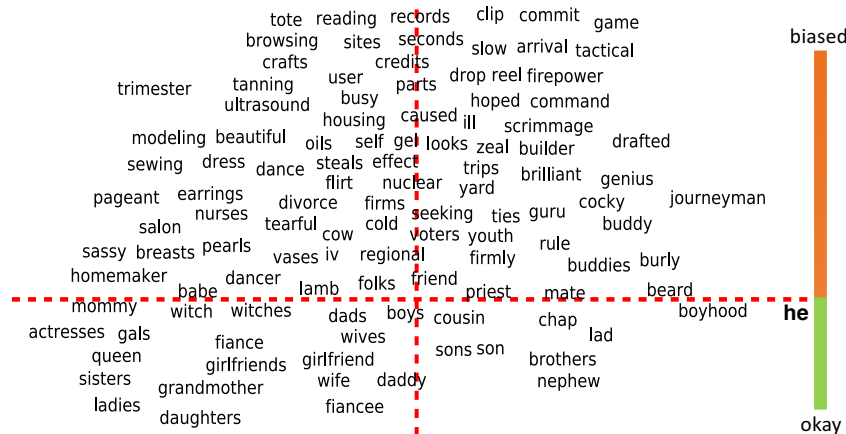

Figure 3: Selected words projected along two axes: $x$ is a projection onto the difference between the embeddings of the words *he* and *she*, and $y$ is a direction learned in the embedding that captures gender neutrality, with gender neutral words above the line and gender specific words below the line. Our hard debiasing algorithm removes the gender pair associations for gender neutral words. In this figure, the words above the horizontal line would all be collapsed to the vertical line.

The first step, called **Identify gender subspace**, is to identify a direction (or, more generally, a subspace) of the embedding that captures the bias. For the second step, we define two options: **Neutralize and Equalize** or **Soften**. **Neutralize** ensures that gender neutral words are zero in the gender subspace. **Equalize** perfectly equalizes sets of words outside the subspace and thereby enforces the property that any neutral word is equidistant to all words in each equality set. For instance, if {grandmother, grandfather} and {guy, gal} were two equality sets, then after equalization *babysit* would be equidistant to *grandmother* and *grandfather* and also equidistant to *gal* and *guy*, but presumably closer to the grandparents and further from the *gal* and *guy*. This is suitable for applications where one does not want any such pair to display any bias with respect to neutral words.

The disadvantage of Equalize is that it removes certain distinctions that are valuable in certain applications. For instance, one may wish a language model to assign a higher probability to the phrase *to grandfather a regulation*) than *to grandmother a regulation* since *grandfather* has a meaning that *grandmother* does not – equalizing the two removes this distinction. The Soften algorithm reduces the differences between these sets while maintaining as much similarity to the original embedding as possible, with a parameter that controls this trade-off.

To define the algorithms, it will be convenient to introduce some further notation. A subspace $B$ is defined by $k$ orthogonal unit vectors $B = \{b_1, \ldots, b_k\} \subset \mathbb{R}^d$. In the case $k = 1$, the subspace is simply a direction. We denote the projection of a vector $v$ onto $B$ by, $v_B = \sum_{j=1}^k (v \cdot b_j) b_j$. This also means that $v - v_B$ is the projection onto the orthogonal subspace.

**Step 1: Identify gender subspace.** Inputs: word sets $W$, defining sets $D_1, D_2, \ldots, D_n \subset W$ as well as embedding $\{\vec{w} \in \mathbb{R}^d\}_{w \in W}$ and integer parameter $k \geq 1$. Let $\mu_i := \sum_{w \in D_i} \vec{w}/|D_i|$ be the means of the defining sets. Let the bias subspace $B$ be the first $k$ rows of $\mathrm{SVD}(\mathbf{C})$ where $\mathbf{C} := \sum_{i=1}^n \sum_{w \in D_i} (\vec{w} - \mu_i)^T (\vec{w} - \mu_i)/|D_i|$.

**Step 2a: Hard de-biasing (neutralize and equalize).** Additional inputs: words to neutralize $N \subseteq W$, family of equality sets $\mathcal{E} = \{E_1, E_2, \ldots, E_m\}$ where each $E_i \subseteq W$. For each word $w \in N$, let $\vec{w}$ be re-embedded to $\vec{w} := (\vec{w} - \vec{w}_B)/\|\vec{w} - \vec{w}_B\|$. For each set $E \in \mathcal{E}$, let

$\mu := \sum_{w \in E} w/|E|$ and $\nu := \mu - \mu_B$. For each $w \in E$, $\vec{w} := \nu + \sqrt{1 - \|\nu\|^2} \frac{\vec{w}_B - \mu_B}{\|\vec{w}_B - \mu_B\|}$. Finally, output the subspace $B$ and the new embedding $\{\vec{w} \in \mathbb{R}^d\}_{w \in W}$.

Equalize equates each set of words outside of $B$ to their simple average $\nu$ and then adjusts vectors so that they are unit length. It is perhaps easiest to understand by thinking separately of the two components $\vec{w}_B$ and $\vec{w}_{\perp B} = \vec{w} - \vec{w}_B$. The latter $\vec{w}_{\perp B}$ are all simply equated to their average. Within $B$, they are centered (moved to mean 0) and then scaled so that each $\vec{w}$ is unit length. To motivate why we center, beyond the fact that it is common in machine learning, consider the bias direction being the gender direction ($k = 1$) and a gender pair such as $E = \{male, female\}$. As discussed, it

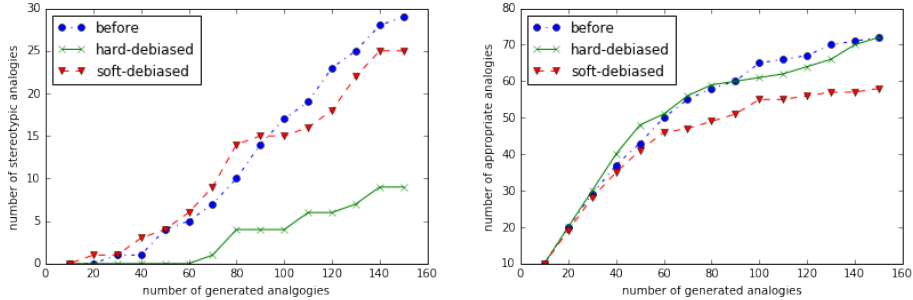

Figure 4: Number of stereotypical (Left) and appropriate (Right) analogies generated by word embeddings before and after debiasing.

so happens that both words are positive (female) in the gender direction, though *female* has a greater projection. One can only speculate as to why this is the case, e.g., perhaps the frequency of text such as *male nurse* or *male escort* or *she was assaulted by the male*. However, because *female* has a greater gender component, after centering the two will be symmetrically balanced across the origin. If instead, we simply scaled each vector's component in the bias direciton without centering, *male* and *female* would have exactly the same embedding and we would lose analogies such as *father:male :: mother:female*. We note that Neutralizing and Equalizing completely remove pair bias.

**Observation 1.** *After Steps 1 and 2a, for any gender neutral word $w$ any equality set $E$, and any two words $e_1, e_2 \in E$, $\vec{w} \cdot \vec{e}_1 = w \cdot \vec{e}_2$ and $\|\vec{w} - \vec{e}_1\| = \|\vec{w} - \vec{e}_2\|$. Furthermore, if $\mathcal{E} = \{\{x, y\} | (x, y) \in P\}$ are the sets of pairs defining PairBias, then* $\text{PairBias} = 0$.

**Step 2b: Soft bias correction**. Overloading the notation, we let $W \in \mathbb{R}^{d \times |vocab|}$ denote the matrix of all embedding vectors and $N$ denote the matrix of the embedding vectors corresponding to gender neutral words. $W$ and $N$ are learned from some corpus and are inputs to the algorithm. The desired debiasing transformation $T \in \mathbb{R}^{d \times d}$ is a linear transformation that seeks to preserve pairwise inner products between all the word vectors while minimizing the projection of the gender neutral words onto the gender subspace. This can be formalized as $\min_T \|(TW)^T(TW) - W^T W\|_F^2 + \lambda \|(TN)^T(TB)\|_F^2$, where $B$ is the gender subspace learned in Step 1 and $\lambda$ is a tuning parameter that balances the objective of preserving the original embedding inner products with the goal of reducing gender bias. For $\lambda$ large, $T$ would remove the projection onto $B$ from all the vectors in $N$, which corresponds exactly to Step 2a. In the experiment, we use $\lambda = 0.2$. The optimization problem is a semi-definite program and can be solved efficiently. The output embedding is normalized to have unit length, $\hat{W} = \{Tw / \|Tw\|_2, w \in W\}$.

## 5 Determining gender neutral words

For practical purposes, since there are many fewer gender specific words, it is more efficient to enumerate the set of gender specific words $S$ and take the gender neutral words to be the compliment, $N = W \setminus S$. Using dictionary definitions, we derive a subset $S_0$ of 218 words out of the words in w2vNEWS. Recall that this embedding is a subset of 26,377 words out of the full 3 million words in the embedding, as described in Section 2. This base list $S_0$ is given in Appendix F. Note that the choice of words is subjective and ideally should be customized to the application at hand.

We generalize this list to the entire 3 million words in the Google News embedding using a linear classifier, resulting in the set $S$ of 6,449 gender-specific words. More specifically, we trained a linear Support Vector Machine (SVM) with regularization parameter of $C = 1.0$. We then ran this classifier on the remaining words, taking $S = S_0 \cup S_1$, where $S_1$ are the words labeled as gender specific by our classifier among the words in the entire embedding that are not in the 26,377 words of w2vNEWS. Using 10-fold cross-validation to evaluate the accuracy, we find an $F$-score of $.627 \pm .102$.

Figure 3 illustrates the results of the classifier for separating gender-specific words from gender-neutral words. To make the figure legible, we show a subset of the words. The $x$-axis correspond to projection of words onto the $\overrightarrow{she} - \overrightarrow{he}$ direction and the $y$-axis corresponds to the distance from the decision boundary of the trained SVM.

# 6 Debiasing results

We evaluated our debiasing algorithms to ensure that they preserve the desirable properties of the original embedding while reducing both direct and indirect gender biases. First we used the same analogy generation task as before: for both the hard-debiased and the soft-debiased embeddings, we automatically generated pairs of words that are analogous to *she-he* and asked crowd-workers to evaluate whether these pairs reflect gender stereotypes. Figure 4 shows the results. On the initial w2vNEWS embedding, 19% of the top 150 analogies were judged as showing gender stereotypes by a majority of the ten workers. After applying our hard debiasing algorithm, only 6% of the new embedding were judged as stereotypical.

As an example, consider the analogy puzzle, *he* to *doctor* is as *she* to $X$. The original embedding returns $X = nurse$ while the hard-debiased embedding finds $X = physician$. Moreover the hard-debiasing algorithm preserved gender appropriate analogies such as *she* to *ovarian cancer* is as *he* to *prostate cancer*. This demonstrates that the hard-debiasing has effectively reduced the gender stereotypes in the word embedding. Figure 4 also shows that the number of appropriate analogies remains similar as in the original embedding after executing hard-debiasing. This demonstrates that that the quality of the embeddings is preserved. The details results are in Appendix J. Soft-debiasing was less effective in removing gender bias. To further confirms the quality of embeddings after debiasing, we tested the debiased embedding on several standard benchmarks that measure whether related words have similar embeddings as well as how well the embedding performs in analogy tasks. Appendix Table 2 shows the results on the original and the new embeddings and the transformation does not negatively impact the performance. In Appendix A, we show how our algorithm also reduces *indirect* gender bias.

# 7 Discussion

Word embeddings help us further our understanding of bias in language. We find a single direction that largely captures gender, that helps us capture associations between gender neutral words and gender as well as indirect inequality. The projection of gender neutral words on this direction enables us to quantify their degree of female- or male-bias.

To reduce the bias in an embedding, we change the embeddings of gender neutral words, by removing their gender associations. For instance, *nurse* is moved to to be equally male and female in the direction $g$. In addition, we find that gender-specific words have additional biases beyond $g$. For instance, *grandmother* and *grandfather* are both closer to *wisdom* than *gal* and *guy* are, which does not reflect a gender difference. On the other hand, the fact that *babysit* is so much closer to *grandmother* than *grandfather* (more than for other gender pairs) is a gender bias specific to *grandmother*. By equating *grandmother* and *grandfather* outside of gender, and since we've removed $g$ from *babysit*, both *grandmother* and *grandfather* and equally close to *babysit* after debiasing. By retaining the gender component for gender-specific words, we maintain analogies such as *she:grandmother :: he:grandfather*. Through empirical evaluations, we show that our hard-debiasing algorithm significantly reduces both direct and indirect gender bias while preserving the utility of the embedding. We have also developed a soft-embedding algorithm which balances reducing bias with preserving the original distances, and could be appropriate in specific settings.

One perspective on bias in word embeddings is that it merely reflects bias in society, and therefore one should attempt to debias society rather than word embeddings. However, by reducing the bias in today's computer systems (or at least not amplifying the bias), which is increasingly reliant on word embeddings, in a small way debiased word embeddings can hopefully contribute to reducing gender bias in society. At the very least, machine learning should not be used to inadvertently amplify these biases, as we have seen can naturally happen.

In specific applications, one might argue that gender biases in the embedding (e.g. *computer programmer* is closer to *he*) could capture useful statistics and that, in these special cases, the original biased embeddings could be used. However given the potential risk of having machine learning algorithms that amplify gender stereotypes and discriminations, we recommend that we should err on the side of neutrality and use the debiased embeddings provided here as much as possible.

**Acknowledgments.** The authors thank Tarleton Gillespie and Nancy Baym for numerous helpful discussions.[5]

## Footnotes

[1] Stereotypes are biases that are widely held among a group of people. We show that the biases in the word embedding are in fact closely aligned with social conception of gender stereotype, as evaluated by U.S.-based crowd workers on Amazon's Mechanical Turk. The crowd agreed that the biases reflected both in the location of vectors (e.g. $\overrightarrow{doctor}$ closer to $\overrightarrow{man}$ than to $\overrightarrow{woman}$) as well as in analogies (e.g., *he*:coward :: *she*:whore.) exhibit common gender stereotypes.

[2]We will abuse terminology and refer to the embedding of a word and the word interchangeably. For example, the statement *cat* is more similar to *dog* than to *cow* means $\overrightarrow{cat} \cdot \overrightarrow{dog} \geq \overrightarrow{cat} \cdot \overrightarrow{cow}$.

[3]https://code.google.com/archive/p/word2vec/

[4]All human experiments were performed on the Amazon Mechanical Turk platform. We selected for U.S.-based workers to maintain homogeneity and reproducibility to the extent possible with crowdsourcing.

[5] This material is based upon work supported in part by NSF Grants CNS-1330008, CCF-1527618, by ONR Grant 50202168, NGA Grant HM1582-09-1-0037 and DHS 2013-ST-061-ED0001

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
