[Supplementary Material · CRv3_supp.pdf]

# A  Indirect bias

**Indirect gender bias.**  The direct bias analyzed above manifests in the relative similarities between gender-specific words and gender neutral words. Gender bias could also affect the relative geometry between gender neutral words themselves. To test this *indirect* gender bias, we take pairs of words that are gender-neutral, for example *softball* and *football*. We project all the occupation words onto the $\overrightarrow{softball} - \overrightarrow{football}$ direction and looked at the extremes words, which are listed in Figure 5. For instance, the fact that the words *bookkeeper* and *receptionist* are much closer to *softball* than *football* may result indirectly from female associations with *bookkeeper*, *receptionist* and *softball*. It's important to point out that that many pairs of male-biased (or female-biased) words have legitimate associations having nothing to do with gender. For example, while both *footballer* and *football* have strong male biases, their similarity is justified by factors other than gender. In Section 3, we define a metric to more rigorously quantify these indirect effects of gender bias.

Unfortunately, the above definitions still do not capture indirect bias. To see this, imagine completely removing from the embedding both words in gender pairs (as well as words such as *beard* or *uterus* that are arguably gender-specific but which cannot be paired). There would still be indirect gender association in that a word that should be gender neutral, such as *receptionist*, is closer to *softball* than *football* (see Figure 5). As discussed in the Introduction, it can be subtle to obtain the ground truth of the extent to which such similarities is due to gender.

The gender subspace $g$ that we have identified allows us to quantify the contribution of $g$ to the similarities between any pair of words. We can decompose a given word vector $w \in \mathbb{R}^d$ as $w = w_g + w_\perp$, where $w_g = (w \cdot g)g$ is the contribution from gender and $w_\perp = w - w_g$. Note that all the word vectors are normalized to have unit length. We define the gender component to the similarity between two word vectors $w$ and $v$ as $\beta(w, v) = \left( w \cdot v - \frac{w_\perp \cdot v_\perp}{\|w_\perp\|_2 \|v_\perp\|_2} \right) \Big/ w \cdot v$.

The intuition behind this metric is as follow: $\frac{w_\perp \cdot v_\perp}{\|w_\perp\|_2 \|v_\perp\|_2}$ is the inner product between the two vectors if we project out the gender subspace and renormalize the vectors to be of unit length. The metric quantifies how much this inner product changes (as a fraction of the original inner product value) due to this operation of removing the gender subspace. Because of noise in the data, every vector has some non-zero component $w_\perp$ and $\beta$ is well-defined. Note that $\beta(w, w) = 0$, which is reasonable since the similarity of a word to itself should not depend on gender contribution. If $w_g = 0 = v_g$, then $\beta(w, v) = 0$; and if $w_\perp = 0 = v_\perp$, then $\beta(w, v) = 1$.

In Figure 5, as a case study, we examine the most extreme words on the $\overrightarrow{softball} - \overrightarrow{football}$ direction. The five most extreme words (i.e. words with the highest positive or the lowest negative projections onto $\overrightarrow{softball} - \overrightarrow{football}$) are shown in the table. Words such as *receptionist*, *waitress* and *homemaker* are closer to *softball* than *football*, and the $\beta$'s between these words and *softball* is substantial (67%, 35%, 38%, respectively). This suggests that the apparent similarity in the embeddings of these words to *softball* can be largely explained by gender biases in the embedding. Similarly, *businessman* and *maestro* are closer to *football* and this can also be attributed largely to indirect gender bias, with $\beta$'s of 31% and 42%, respectively.

**Effect of debiasing algorithm on indirect bias.**  We also investigated how the strict debiasing algorithm affects indirect gender bias. Because we do not have the ground truth on the indirect effects of gender bias, it is challenging to quantify the performance of the algorithm in this regard. However we do see promising qualitative improvements, as shown in Figure 5 in the *softball*, *football* example. After applying the strict debias algorithm, we repeated the experiment and show the most extreme words in the $\overrightarrow{softball} - \overrightarrow{football}$ direction. The most extreme words closer to *softball* are now *infielder* and *major leaguer* in addition to *pitcher*, which are more relevant and do not exhibit gender bias. Gender stereotypic associations such are *receptionist*, *waitress* and *homemaker* are moved down the list. Similarly, words that clearly show male bias, e.g. *businessman*, are also no longer at the top of the list. Note that the two most extreme words in the $\overrightarrow{softball} - \overrightarrow{football}$ direction are *pitcher* and *footballer*. The similarities between *pitcher* and *softball* and between *footballer* and *football* comes from the actual functions of these words and hence have little gender contribution. These two words are essentially unchanged by the debiasing algorithm.

| *softball* **extreme** | **gender portion** | **after debiasing** |
|---|---|---|
| 1. pitcher | -1% | 1. pitcher |
| 2. bookkeeper | 20% | 2. infielder |
| 3. receptionist | 67% | 3. major leaguer |
| 4. registered nurse | 29% | 4. bookkeeper |
| 5. waitress | 35% | 5. investigator |

| *football* **extreme** | **gender portion** | **after debiasing** |
|---|---|---|
| 1. footballer | 2% | 1. footballer |
| 2. businessman | 31% | 2. cleric |
| 3. pundit | 10% | 3. vice chancellor |
| 4. maestro | 42% | 4. lecturer |
| 5. cleric | 2% | 5. midfielder |

Figure 5: **Example of indirect bias**. The five most extreme occupations on the *softball-football* axis, which indirectly captures gender bias. For each occupation, the degree to which the association represents a gender bias is shown, as described in Section A.

Figure 6: Comparing the bias of two different embeddings–the w2vNEWS and the GloVe web-crawl embedding. In each embedding, the occupation words are projected onto the *she-he* direction. Each dot corresponds to one occupation word; the gender bias of occupations is highly consistent across embeddings (Spearman $\rho = 0.81$).

## B    Consistency of gender stereotypes across embeddings

In Figure 6, we studied the gender stereotype on GloVe word embeddings trained on a courpus of web-crawl texts. The gender bias of occupational words here are highly consistent with the bias in the w2vNEWS embeddings.

## C    Generating analogies

We now expand on different possible methods for generating $(x, y)$ pairs, given $(a, b)$ for generating analogies $a{:}x :: b{:}y$. The first and simplest metric is to consider scoring an analogy by $\|(\vec{a}-\vec{b})-(\vec{x}-\vec{y})\|$. This may be called the *parallelogram* approach and, for the purpose of finding the best single $y$ given $a, b, x$, it is equivalent to the most common approach to finding single word analogies, namely maximizing $\cos(\vec{y}, \vec{x} + \vec{b} - \vec{a})$ called *cosAdd* in earlier work [21] since we assume all vectors are unit length. This works well in some cases, but a weakness can be seen that, for many triples $(a, b, x)$, the closest word to $x$ is $y = x$, i.e., $x = \arg\min_y \|(\vec{a} - \vec{b}) - (\vec{x} - \vec{y})\|$. As a result, the definition explicitly excludes the possibility of returning $x$ itself. In these cases, $y$ is often a word very similar to $x$, and in most of these cases such an algorithm produces two opposing analogies: $a{:}x :: b{:}y$ as well as $a{:}y :: b{:}x$, which violates a desideratum of analogies (see [33], section 2.2).

| Analogies generated using eq. (2) | Analogies generated using our approach, eq. (1) |
|---|---|
| petite-diminutive | petite-lanky |
| seventh inning-eighth inning | volleyball-football |
| seventh-sixth | interior designer-architect |
| east-west | bitch-bastard |
| tripled-doubled | bra-pants |
| breast cancer-cancer | nurse-surgeon |
| meter hurdles-meter dash | feminine-manly |
| thousands-tens | glamorous-flashy |
| eight-seven | registered nurse-physician |
| unemployment rate-jobless rate | cupcakes-pizzas |

Figure 7: First 10 different *she-he* analogies generated using the parallelogram approach and our approach, from the top 100 *she-he* analogies not containing gender specific words. Most of the analogies on the left seem to have little connection to gender.

Related issues are discussed in [33, 18], the latter of which proposes the 3CosMul objective to finding $y$ given $(a, b, x)$:

$$\max_y \frac{(1 + \cos(\vec{x}, \vec{y}))(1 + \cos(\vec{x}, \vec{b}))}{1 + \cos(\vec{y}, \vec{a}) + \epsilon}.$$

The additional $\epsilon$ is necessary so that the denominator is positive. This approach is designed for finding a single word $y$ and not directly applicable for the problem of generating both $x$ and $y$ as the objective is not symmetric in $x$ and $y$.

In the spirit of their work, we note that a desired property is that the direction $\vec{a} - \vec{b}$ should be similar (in angle) to the direction $\vec{x} - \vec{y}$ even if the magnitudes differ. Interestingly, given $(a, b, x)$, the $y$ that maximizes $\cos(\vec{a} - \vec{b}, \vec{x} - \vec{y})$ is generally an extreme. For instance, for $a =$*he and* $b =$she, for the vast majority of words $x$, the word *her* maximizes the expression for $y$. This is due to the fact that the most significant difference between a random word $x$ and the word *her* is that *her* is likely much more feminine than $x$. Since, from a perceptual point of view it is easier to compare and contrast similar items than very different items, we instead seek $x$ and $y$ that are not semantically similar, which is why our definition is restricted to $\|\vec{x} - \vec{y}\| \le \delta$.

As $\delta$ varies from small to large, the analogies vary from generating very similar $x$ and $y$ to very loosely related $x$ and $y$ where their relationship is vague and more "creative".

Finally, Figure 7 highlights differences between analogies generated from our approach and the corresponding analogies generated by the first approach mentioned above, namely minimizing:

$$\min_{x, y: x \ne a, y \ne b, x \ne y} \|(\vec{a} - \vec{b}) - (\vec{x} - \vec{y})\|, \tag{2}$$

To compare, we took the first 100 analogies generated using the two approaches that did not have any gender-specific words. We then display the first 10 analogies from each list which do not occur in the other list of 100.

## D  Learning the linear transform

In the soft debiasing algorithm, we need to solve the following optimization problem.

$$\min_T \|(TW)^T(TW) - W^TW\|_F^2 + \lambda \|(TN)^T(TB)\|_F^2.$$

Let $X = T^TT$, then this is equivalent to the following semi-definite programming problem

$$\min_X \|W^TXW - W^TW\|_F^2 + \lambda \|N^TXB\|_F^2 \qquad \text{s.t.} X \succeq 0. \tag{3}$$

The first term ensures that the pairwise inner products are preserved and the second term induces the biases of gender neutral words onto the gender subspace to be small. The user-specified parameter $\lambda$ balances the two terms.

|             | RG   | WS   | analogy |
|-------------|------|------|---------|
| Before      | 76.1 | 70.0 | 71.2    |
| Hard-debiased | 76.5 | 69.7 | 71.2    |
| Soft-debiased | 76.9 | 69.7 | 71.2    |

Table 1: The columns show the performance of the original, complete w2vNEWS embedding ("before") and the debiased w2vNEWS on the standard evaluation metrics measuring coherence and analogy-solving abilities: RG [27], WS [10], MSR-analogy [21]. Higher is better. The results show that the performance does not degrade after debiasing.

Directly solving this SDP optimization problem is challenging. In practice, the dimension of matrix $W$ is in the scale of $300 \times 400,000$. The dimensions of the matrices $W^T X W$ and $W^T W$ are $400,000 \times 400,000$, causing computational and memory issues. We perform singular value decomposition on $W$, such that $W = U \Sigma V^T$, where $U$ and $V$ are orthogonal matrices and $\Sigma$ is a diagonal matrix.

$$
\begin{aligned}
\|W^T X W - W^T W\|_F^2 &= \|W^T (X - I) W\|_F^2 \\
&= \|V \Sigma U^T (X - I) U \Sigma V^T\|_F^2 \\
&= \|\Sigma U^T (X - I) U \Sigma\|_F^2.
\end{aligned}
\tag{4}
$$

The last equality follows the fact that $V$ is an orthogonal matrix and ($\|VYV^T\|_F^2 = tr(VY^T V^T V Y V^T) = tr(VY^T Y V^T) = tr(Y^T Y V^T V) = tr(Y^T Y) = \|Y\|_F^2$.)

Substituting Eq. (4) to Eq. (3) gives

$$
\min_X \|\Sigma U^T (X - I) U \Sigma\|_F^2 + \lambda \|PXS^T\|_F^2 \qquad \text{s.t. } X \succeq 0.
\tag{5}
$$

Here $\Sigma U^T (X - I) U \Sigma$ is a $300 \times 300$ matrix and can be solved efficiently. The solution $T$ is the debiasing transformation of the word embedding.

## E  Debiasing the full w2vNEWS embedding.

In the main text, we focused on the results from a cleaned version of w2vNEWS consisting of 26,377 lower-case words. We have also applied our hard debiasing algorithm to the full w2vNEWS dataset. Evalution based on the standard metrics shows that the debiasing does not degrade the utility of the embedding (Table 1).

## F  Details of gender specific words base set

This section gives precise details of how we derived our list of gender neutral words. Note that the choice of gender neutral words is partly subjective. Some words are most often associated with females or males but have exceptions, such as *beard* (bearded women), *estrogen* (men have small amounts of the hormone estrogen), and *rabbi* (reformed Jewish congregations recognize female rabbis). There are also many words that have multiple senses, some of which are gender neutral and others of which are gender specific. For instance, the profession of *nursing* is gender neutral while *nursing* a baby (i.e., breastfeeding) is only performed by women.

To derive the base subset of words from w2vNEWS, for each of the 26,377 words in the filtered embedding, we selected words whose definitions include any of the following words in their singular or plural forms: *female, male, woman, man, girl, boy, sister, brother, daughter, son, grandmother, grandfather, wife, husband*. Definitions were taken from Wordnet [9] (in the case where a word had multiple senses/synsets, we chose the definition whose corresponding lemma had greatest frequency in terms of its count). This list of hundreds of words contains most gender specific words of interest but also contains some gender neutral words, e.g., the definition of *mating* is "the act of pairing a male and female for reproductive purposes." Even though the word *female* is in the definition, *mating* is not gender specific. We went through this list and manually selected those words that were clearly gender specific. Motivated by the application of improving web search, we used a strict definition

of gender specificity, so that when in doubt a word was defined to be gender neutral. For instance, clothing words (e.g., the definition of *vest* is "a collarless men's undergarment for the upper part of the body") were classified as gender neutral since there are undoubtedly people of every gender that wear any given type of clothing. After this filtering, we were left with the following list of 218 gender-specific words (sorted by word frequency):

*he, his, her, she, him, man, women, men, woman, spokesman, wife, himself, son, mother, father, chairman, daughter, husband, guy, girls, girl, boy, boys, brother, spokeswoman, female, sister, male, herself, brothers, dad, actress, mom, sons, girlfriend, daughters, lady, boyfriend, sisters, mothers, king, businessman, grandmother, grandfather, deer, ladies, uncle, males, congressman, grandson, bull, queen, businessmen, wives, widow, nephew, bride, females, aunt, prostate cancer, lesbian, chairwoman, fathers, moms, maiden, granddaughter, younger brother, lads, lion, gentleman, fraternity, bachelor, niece, bulls, husbands, prince, colt, salesman, hers, dude, beard, filly, princess, lesbians, councilman, actresses, gentlemen, stepfather, monks, ex girlfriend, lad, sperm, testosterone, nephews, maid, daddy, mare, fiance, fiancee, kings, dads, waitress, maternal, heroine, nieces, girlfriends, sir, stud, mistress, lions, estranged wife, womb, grandma, maternity, estrogen, ex boyfriend, widows, gelding, diva, teenage girls, nuns, czar, ovarian cancer, countrymen, teenage girl, penis, bloke, nun, brides, housewife, spokesmen, suitors, menopause, monastery, motherhood, brethren, stepmother, prostate, hostess, twin brother, schoolboy, brotherhood, fillies, stepson, congresswoman, uncles, witch, monk, viagra, paternity, suitor, sorority, macho, businesswoman, eldest son, gal, statesman, schoolgirl, fathered, goddess, hubby, stepdaughter, blokes, dudes, strongman, uterus, grandsons, studs, mama, godfather, hens, hen, mommy, estranged husband, elder brother, boyhood, baritone, grandmothers, grandpa, boyfriends, feminism, countryman, stallion, heiress, queens, witches, aunts, semen, fella, granddaughters, chap, widower, salesmen, convent, vagina, beau, beards, handyman, twin sister, maids, gals, housewives, horsemen, obstetrics, fatherhood, councilwoman, princes, matriarch, colts, ma, fraternities, pa, fellas, councilmen, dowry, barbershop, fraternal, ballerina*

## G   Questionnaire for generating gender stereotypical words

**Task: for each category, please enter 10 or more words, separated by commas.** We are looking for a variety of creative answers – this is a mentally challenging HIT that will make you think.

- **10 or more comma-separated words definitionally associated with males.**
  Examples: *dude, menswear, king, penis*, ...
- **10 or more comma-separated words definitionally associated with females.**
  Examples: *queen, Jane, girl*, ...
- **10 or more comma-separated words stereotypically associated with males**
  Examples: *football, janitor, cocky*, ...
- **10 or more comma-separated words stereotypically associated with females**
  Examples: *pink, sewing, caring, sassy, nurse*, ...

Thank you for your help in making Artificially Intelligent systems that aren't prejudiced. :-)

## H   Questionnaire for generating gender stereotypical analogies

An analogy describes two pairs of words where the relationship between the two words in each pair is the same. An example of an analogy is *apple* is to *fruit* as *asparagus* is to *vegetable* (denoted as apple:fruit::asparagus:vegetable). We need your help to improve our analogy generating system.

   **Task: please enter 10 or more analogies reflecting gender stereotypes, separated by commas**. We are looking for a variety of creative answers – this is a mentally challenging HIT that will make you think.

**Examples of stereotypes**

- tall : man :: short : woman reflects a cultural stereotype that men are tall and women are short.

- doctor : man :: nurse : woman reflects a stereotype that doctors are typically men and nurses are typically women.

## I    Questionnaire for rating stereotypical analogies

An analogy describes two pairs of words where the relationship between the two words in each pair is the same. An example of an analogy is *apple* is to *fruit* as *asparagus* is to *vegetable* (denoted as apple:fruit::asparagus:vegetable). We need your help to improve our analogy generating system.

**Task: Which analogies are stereotypes? Which ones are appropriate analogies?**

- **Examples of stereotype analogies**
  tall : man :: short : woman
  doctor : man :: nurse : woman
- **Examples of appropriate analogies**
  King: man :: Queen : woman
  brother : man :: sister : woman
  John : man :: Mary : woman
  His : man :: Hers : woman
  salesman : man :: saleswoman : woman
  penis : man :: vagina : woman

WARNING: This HIT may contain adult content. Worker discretion is advised.

Check the analogies that are stereotypes

...

Check the analogies that are nonsensical

...

Check the analogies that are nonsensical

...

Any suggestions or comments on the hit? Optional feedback

## J    Analogies Generated by Word Embeddings

| After executing hard debiasing | | | Before executing debiasing | | |
|---|---|---|---|---|---|
| Analogy | Appropriate | Biased | Analogy | Appropriate | Biased |
| hostess:bartender | 1 | 8 | midwife:doctor | 1 | 10 |
| ballerina:dancer | 0 | 7 | sewing:carpentry | 2 | 9 |
| colts:mares | 6 | 7 | pediatrician:orthopedic_surgeon | 0 | 9 |
| ma:na | 8 | 7 | registered_nurse:physician | 1 | 9 |
| salesperson:salesman | 1 | 7 | housewife:shopkeeper | 1 | 9 |
| diva:superstar | 4 | 7 | skirts:shorts | 0 | 9 |
| witches:vampires | 1 | 7 | nurse:surgeon | 1 | 9 |
| hair_salon:barbershop | 4 | 6 | interior_designer:architect | 1 | 8 |
| maid:housekeeper | 3 | 6 | softball:baseball | 4 | 8 |
| soprano:baritone | 4 | 5 | blond:burly | 2 | 8 |
| footy:blokes | 0 | 5 | nanny:chauffeur | 1 | 8 |
| maids:servants | 4 | 5 | feminism:conservatism | 2 | 8 |
| dictator:strongman | 0 | 5 | adorable:goofy | 0 | 8 |
| bachelor:bachelor_degree | 7 | 4 | vocalists:guitarists | 0 | 8 |
| witch:witchcraft | 0 | 4 | cosmetics:pharmaceuticals | 1 | 8 |
| gaffer:lads | 1 | 3 | whore:coward | 0 | 7 |
| convent:monastery | 8 | 3 | vocalist:guitarist | 1 | 7 |

| Pair | | | Pair | | |
|---|---|---|---|---|---|
| hen:cock | 8 | 2 | petite:lanky | 1 | 7 |
| aldermen:councilmen | 0 | 2 | salesperson:salesman | 1 | 7 |
| girlfriend:friend | 0 | 2 | sassy:snappy | 2 | 7 |
| housewife:homemaker | 2 | 2 | diva:superstar | 4 | 7 |
| maternal:infant_mortality | 1 | 2 | charming:affable | 2 | 6 |
| beau:lover | 1 | 2 | giggle:chuckle | 1 | 6 |
| mistress:prostitute | 0 | 2 | witch:demon | 2 | 6 |
| heroine:protagonist | 2 | 2 | volleyball:football | 1 | 6 |
| heiress:socialite | 2 | 2 | feisty:mild_mannered | 0 | 6 |
| teenage_girl:teenager | 3 | 2 | cupcakes:pizzas | 1 | 6 |
| estrogen:testosterone | 9 | 2 | dolls:replicas | 0 | 6 |
| actresses:actors | 10 | 1 | netball:rugby | 0 | 6 |
| blokes:bloke | 1 | 1 | hairdresser:barber | 6 | 5 |
| girlfriends:buddies | 6 | 1 | soprano:baritone | 4 | 5 |
| compatriot:countryman | 3 | 1 | gown:blazer | 6 | 5 |
| compatriots:countrymen | 2 | 1 | glamorous:flashy | 2 | 5 |
| gals:dudes | 10 | 1 | sweater:jersey | 0 | 5 |
| eldest:elder_brother | 1 | 1 | feminist:liberal | 0 | 5 |
| sperm:embryos | 2 | 1 | bra:pants | 2 | 5 |
| mother:father | 10 | 1 | rebounder:playmaker | 0 | 5 |
| wedlock:fathered | 0 | 1 | nude:shirtless | 0 | 5 |
| mama:fella | 7 | 1 | judgmental:arrogant | 1 | 4 |
| lesbian:gay | 8 | 1 | boobs:ass | 1 | 4 |
| kid:guy | 1 | 1 | salon:barbershop | 7 | 4 |
| carpenter:handyman | 5 | 1 | lovely:brilliant | 0 | 4 |
| she:he | 9 | 1 | practicality:durability | 0 | 4 |
| herself:himself | 10 | 1 | singer:frontman | 0 | 4 |
| her:his | 10 | 1 | gorgeous:magnificent | 2 | 4 |
| uterus:intestine | 1 | 1 | ponytail:mustache | 2 | 4 |
| queens:kings | 10 | 1 | feminists:socialists | 0 | 4 |
| female:male | 9 | 1 | bras:trousers | 5 | 4 |
| women:men | 10 | 1 | wedding_dress:tuxedo | 6 | 4 |
| pa:mo | 9 | 1 | violinist:virtuoso | 0 | 4 |
| nun:monk | 7 | 1 | handbag:briefcase | 8 | 3 |
| matriarch:patriarch | 9 | 1 | giggling:grinning | 0 | 3 |
| nuns:priests | 9 | 1 | kids:guys | 3 | 3 |
| menopause:puberty | 2 | 1 | beautiful:majestic | 1 | 3 |
| fiance:roommate | 0 | 1 | feminine:manly | 8 | 3 |
| daughter:son | 9 | 1 | convent:monastery | 8 | 3 |
| daughters:sons | 10 | 1 | sexism:racism | 0 | 3 |
| spokeswoman:spokesman | 10 | 1 | pink:red | 0 | 3 |
| politician:statesman | 1 | 1 | blouse:shirt | 6 | 3 |
| stallion:stud | 7 | 1 | bitch:bastard | 8 | 2 |
| suitor:takeover_bid | 8 | 1 | wig:beard | 4 | 2 |
| waitress:waiter | 10 | 1 | hysterical:comical | 0 | 2 |
| lady:waitress | 0 | 1 | male_counterparts:counterparts | 1 | 2 |
| bride:wedding | 0 | 1 | beauty:grandeur | 0 | 2 |
| widower:widowed | 3 | 1 | cheerful:jovial | 0 | 2 |
| husband:younger_brother | 3 | 1 | breast_cancer:lymphoma | 3 | 2 |
| actress:actor | 9 | 0 | heiress:magnate | 6 | 2 |
| mustache:beard | 0 | 0 | estrogen:testosterone | 9 | 2 |
| facial_hair:beards | 0 | 0 | starlet:youngster | 2 | 2 |
| suitors:bidders | 6 | 0 | Mary:John | 9 | 1 |
| girl:boy | 9 | 0 | actresses:actors | 10 | 1 |
| childhood:boyhood | 1 | 0 | middle_aged:bearded | 0 | 1 |
| girls:boys | 10 | 0 | mums:blokes | 5 | 1 |
| counterparts:brethren | 4 | 0 | girlfriends:buddies | 6 | 1 |
| brides:bridal | 1 | 0 | mammogram:colonoscopy | 0 | 1 |
| sister:brother | 10 | 0 | compatriot:countryman | 3 | 1 |

| Pair | | | Pair | | |
|---|---|---|---|---|---|
| friendship:brotherhood | 3 | 0 | luscious:crisp | 0 | 1 |
| sisters:brothers | 9 | 0 | gals:dudes | 10 | 1 |
| businesswoman:businessman | 9 | 0 | siblings:elder_brother | 1 | 1 |
| businesspeople:businessmen | 1 | 0 | mother:father | 10 | 1 |
| chairwoman:chairman | 10 | 0 | babe:fella | 9 | 1 |
| bastard:chap | 0 | 0 | lesbian:gay | 8 | 1 |
| hens:chickens | 3 | 0 | breasts:genitals | 0 | 1 |
| viagra:cialis | 1 | 0 | wonderful:great | 0 | 1 |
| filly:colt | 9 | 0 | she:he | 9 | 1 |
| fillies:colts | 8 | 0 | herself:himself | 10 | 1 |
| congresswoman:congressman | 9 | 0 | her:his | 10 | 1 |
| councilwoman:councilman | 9 | 0 | mommy:kid | 0 | 1 |
| wife:cousin | 0 | 0 | queens:kings | 10 | 1 |
| mom:dad | 10 | 0 | female:male | 9 | 1 |
| mommy:daddy | 10 | 0 | women:men | 10 | 1 |
| moms:dads | 9 | 0 | boyfriend:pal | 0 | 1 |
| widow:deceased | 0 | 0 | matriarch:patriarch | 9 | 1 |
| gal:dude | 9 | 0 | nun:priest | 10 | 1 |
| stepmother:eldest_son | 3 | 0 | breast:prostate | 9 | 1 |
| deer:elk | 1 | 0 | daughter:son | 9 | 1 |
| estranged_husband:estranged | 0 | 0 | daughters:sons | 10 | 1 |
| ex_boyfriend:ex_girlfriend | 7 | 0 | spokeswoman:spokesman | 10 | 1 |
| widows:families | 4 | 0 | fabulous:terrific | 3 | 1 |
| motherhood:fatherhood | 10 | 0 | headscarf:turban | 6 | 1 |
| mothers:fathers | 10 | 0 | waitress:waiter | 10 | 1 |
| guys:fellas | 1 | 0 | husband:younger_brother | 3 | 1 |
| feminism:feminist | 1 | 0 | hers:yours | 2 | 1 |
| womb:fetus | 0 | 0 | teenage_girls:youths | 0 | 1 |
| sorority:fraternity | 9 | 0 | actress:actor | 9 | 0 |
| lesbians:gays | 9 | 0 | blonde:blond | 4 | 0 |
| mare:gelding | 7 | 0 | girl:boy | 9 | 0 |
| fella:gentleman | 1 | 0 | childhood:boyhood | 1 | 0 |
| ladies:gentlemen | 10 | 0 | girls:boys | 10 | 0 |
| boyfriends:girlfriend | 3 | 0 | sister:brother | 10 | 0 |
| goddess:god | 9 | 0 | sisters:brothers | 9 | 0 |
| grandmother:grandfather | 10 | 0 | businesswoman:businessman | 9 | 0 |
| grandma:grandpa | 9 | 0 | chairwoman:chairman | 10 | 0 |
| grandmothers:grandparents | 5 | 0 | filly:colt | 9 | 0 |
| granddaughter:grandson | 10 | 0 | fillies:colts | 8 | 0 |
| granddaughters:grandsons | 9 | 0 | congresswoman:congressman | 9 | 0 |
| me:him | 2 | 0 | councilwoman:councilman | 9 | 0 |
| queen:king | 10 | 0 | mom:dad | 10 | 0 |
| youngster:lad | 1 | 0 | moms:dads | 9 | 0 |
| elephant:lion | 0 | 0 | gal:dude | 9 | 0 |
| elephants:lions | 0 | 0 | motherhood:fatherhood | 10 | 0 |
| manly:macho | 4 | 0 | mothers:fathers | 10 | 0 |
| females:males | 10 | 0 | sorority:fraternity | 9 | 0 |
| woman:man | 8 | 0 | mare:gelding | 7 | 0 |
| fiancee:married | 4 | 0 | lady:gentleman | 9 | 0 |
| maternity:midwives | 1 | 0 | ladies:gentlemen | 10 | 0 |
| monks:monasteries | 0 | 0 | goddess:god | 9 | 0 |
| niece:nephew | 9 | 0 | grandmother:grandfather | 10 | 0 |
| nieces:nephews | 9 | 0 | grandma:grandpa | 9 | 0 |
| hubby:pal | 1 | 0 | granddaughter:grandson | 10 | 0 |
| obstetrics:pediatrics | 3 | 0 | granddaughters:grandsons | 9 | 0 |
| vagina:penis | 10 | 0 | kinda:guy | 1 | 0 |
| princess:prince | 9 | 0 | heroine:hero | 9 | 0 |
| colon:prostate | 6 | 0 | me:him | 2 | 0 |
| ovarian_cancer:prostate_cancer | 10 | 0 | queen:king | 10 | 0 |

| | | | | | |
|---|---|---|---|---|---|
| salespeople:salesmen | 2 | 0 | females:males | 10 | 0 |
| semen:saliva | 7 | 0 | woman:man | 8 | 0 |
| schoolgirl:schoolboy | 8 | 0 | niece:nephew | 9 | 0 |
| replied:sir | 0 | 0 | nieces:nephews | 9 | 0 |
| spokespeople:spokesmen | 0 | 0 | vagina:penis | 10 | 0 |
| boyfriend:stepfather | 1 | 0 | princess:prince | 9 | 0 |
| stepdaughter:stepson | 9 | 0 | ovarian_cancer:prostate_cancer | 10 | 0 |
| teenage_girls:teenagers | 1 | 0 | schoolgirl:schoolboy | 8 | 0 |
| hers:theirs | 0 | 0 | spokespeople:spokesmen | 0 | 0 |
| twin_sister:twin_brother | 9 | 0 | stepdaughter:stepson | 9 | 0 |
| aunt:uncle | 9 | 0 | twin_sister:twin_brother | 9 | 0 |
| aunts:uncles | 10 | 0 | aunt:uncle | 9 | 0 |
| husbands:wives | 7 | 0 | aunts:uncles | 10 | 0 |