[Reviews · NeurIPS 2016]

Reviewer 1

Summary

The paper describes the inherent gender bias in word embeddings, and an algorithm to remove it.

Qualitative Assessment

The paper is well-written (even though it would benefit from a spell-checker and some additional commas) and technically solid, with convincing experimental results. Moreover, it covers a timely and important issue. I would recommend this paper for acceptance at NIPS. If there is anything to change, I would remove the focus on gender stereotypes. You mention a couple of other examples of bias (ethnicity, etc), and it would probably strengthen the paper to take a broader stance. While you say that the number of works on biases is too vast to survey, I was expecting a few more mentions. In particular, I was missing some references to related recent work in NLP and blogs, such as the age bias Hovy and Soegaard (2015) describe, or the post by Ben Schmidt (http://bookworm.benschmidt.org/posts/2015-10-30-rejecting-the-gender-binary.html). The latter also qualifies your claim that you "initiate the study of stereotypes in word embeddings", so I would recommend modifying the claim. There also seems to be an upcoming paper on general biases in NLP: http://www.dirkhovy.com/portfolio/papers/download/ethics.pdf In sum: I would like to see the paper accepted, but recommend inclusion of more related work, and some careful proofreading. Miscellaneous: - The following sentence makes no sense, even after several reads: "We have already well linear directions in embedding capture semantics, in general, and biases, in particular." Missing verb? - duplicated phrases: "and associations between caring and murder while associations between nurturing and murder"

Confidence in this Review

3-Expert (read the paper in detail, know the area, quite certain of my opinion)


Reviewer 2

Summary

This paper deals with stereotypes in word embeddings i.e. by being totally data driven, the embeddings learn stereotypical facts about gender which might be detrimental if the embeddings are to be used in some downstream task like web search ranking. The paper proposes identifying the “gender-stereotype” subspace of the embeddings and eliminating it while doing correction for preserving the other aspects of the embeddings. The results show marked reduction in the aforementioned “sexism” in the embeddings.

Qualitative Assessment

The paper is well written and tackles an interesting problem. The algorithm and the results seem sound to me. However, I have a few concerns regarding the paper: 1). Knocking out the “sexist” subspace and performing “surgery” later leaves several questions answered. How do the performance of these embeddings benchmark against the standard embeddings on standard NLP tasks like NER, Chunking etc.? 2). Relying on crowd-sourcing to get stereotype word pairs and definitional word pairs needs more justification. In the experimental results, the authors mention that “embedding generated 58 pairs and none of it were judged as stereotypical”. Was it the same set of annotators as earlier, who gave the list of stereotype and definitional word pairs? 3). At a meta-level I feel that this approach is quite tailored to U.S based audience. Stereotypes have a huge cultural element to them. For instance, some of the stereotypes described in the paper also exist in Western Europe. It will be helpful to have some annotators from there too, as all the annotators currently were U.S based. 4).Typos: Page 6, “We have already well linear”, “possible while combating”.

Confidence in this Review

2-Confident (read it all; understood it all reasonably well)


Reviewer 3

Summary

This paper proposes an interesting approach to the task of detecting and removing stereotypes in word embeddings. The idea is interesting and the proposed solution appears to be useful. One major issue with the paper is that the proposed solution is rather straightforward, which involves very simple mathematical principles and lacks the technical depth. The technique involves little rigorous mathematical modelling, and makes use of very simple tools such as PCA-based subspace methods. Although the results are interesting, the paper does not appear to be appropriate for NIPS.

Qualitative Assessment

The idea is interesting and the proposed solution seems interesting. However the algorithm appears a little straightforward and does not involve rigorous mathematical justifications.

Confidence in this Review

2-Confident (read it all; understood it all reasonably well)


Reviewer 4

Summary

Word embedding approach is wildly used in many machine learning tasks. Because it is learned from huge collected language dataset like Google News articles, the serious stereotypes may be inherent in the dataset and they typically result in the bias model. The authors initiate the study of stereotypes in word embeddings and try to develop two metrics to quantify gender stereotypes through crowdsourcing. In addition, the authors focus on gender stereotypes and propose an algorithm for learning a transformation to remove it from the embedding while preserving useful properties in a low-dimensional subspace. A experiment is shown to demonstrate that the proposed approach addresses the gender stereotypes issue and does not negatively impact the performance on the standard evaluation metrics measuring coherence and analogy-solving abilities.

Qualitative Assessment

The paper addresses a novel problem which is inherent in the wildly used word embedding model. However, there are indeed something can be improved. -) Although the idea is quite novel, the technical contributions in this work are limited. I wonder that the proposed debiasing algorithm useing PCA to identify the stereotype subspace and a linear transformation to reduce it could work on more complex stereotype problems. -) It lacks the systematic analysis to demonstrate the usefulness of the proposed debiasing algorithm. In other words, it lacks the higher level natural language experiment to prove the effectiveness of the proposed method. -) The analysis for the several hyperparameters such like δ and the threshold of the score S at line 146 and line 151 are missed. -) The number of analogies for experiments are too small. It may increase the number of analogies from other stereotypes (e.g. race, cultural, employment). -) There are several typos in the article line 10 between between -> between line 31 [17] Consider -> [17]. Consider line 33 networks.” -> networks.”. line 163 were judged to -> were judged to. line 231 a a -> a line 246 corresponds corresponds -> corresponds line 278 Table -> Figure2. The article needs to be proofread before submission. -) The readability of the section "Step 3: surgery to modify gender-definition words" is too terrible to follow. It needs to be revised and clarified. I would request the authors to provide response to the questions/explainations as mentioned above.

Confidence in this Review

2-Confident (read it all; understood it all reasonably well)


Reviewer 5

Summary

This paper proposes a pipeline to deal with gender stereotypes in word embeddings, including exploring, quantifying, and debiasing.

Qualitative Assessment

This is an interesting application paper, which addresses the problem of stereotypes in word embeddings. The authors describe the problem well, and the proposed post-processing is well motivated because it seems unlikely to debias gender stereotypes during word embedding learning. Surprisingly, the authors show gender stereotypes largely lie in principle components of embeddings. Such fining may also help to better explain embedding features and is potentially useful to analyzing other aspects of word embeddings. When debiasing, the authors have carefully dealt with gender-related, but not stereotypical words (e.g., man, woman, brother, sister). They first use logistic regression to detect such words. Then (augmented) original embeddings and transformed (but surged) embeddings are weighted summed. Well, the idea of concatenating the original space and surged space makes sense and is sort of similar to Frustratingly easy domain adaptation, ACL-07. I have several comments, which are minor and not major. 1. Regarding the gender direction: I agree the choice of "he-she" is better than "man-woman," but there could be other choices, e.g., "his-her." An average over several pairs may lead to more robust results. 2. The dataset is small, containing only 200 pairs, but the debiasing approach appears to be effective 44.3% --> 0% Very minor: Line 31: Comma after [17] Line 118: a embedding --> an embedding Line 163: not complete Line 246: corresponds corresponds --> corresponds

Confidence in this Review

2-Confident (read it all; understood it all reasonably well)